# Improving End-To-End Latency Fairness Using a Reinforcement-Learning-Based Network Scheduler

**Juhyeok Kwon** [1] , **Jihye Ryu** [1] , **Jee Hang Lee** [1,2] **and Jinoo Joung** [1,2,*]

1    Department of AI & Informatics, Sangmyung University, Seoul 03016, Republic of Korea
2    Department of Human-Centered Artificial Intelligence, Sangmyung University,
     Seoul 03016, Republic of Korea
*    Correspondence: jjoung@smu.ac.kr

**Abstract:** In services such as metaverse, which should provide a constant quality of service (QoS) regardless of the user's physical location, the end-to-end (E2E) latency must be fairly distributed over any flow in the network. To this end, we propose a reinforcement learning (RL)-based scheduler for minimizing the maximum network E2E latency. The RL model used the double deep Q-network (DDQN) with the prioritized experience replay (PER). In order to see the performance change according to the type of RL agent, we implemented a single-agent environment where the controller is an agent and a multi-agent environment where each node is an agent. Since the agents were unable to identify E2E latencies in the multi-agent environment, the state and reward were formulated using the estimated E2E latencies. To precisely evaluate the RL-based scheduler, we set out benchmark algorithms to compare with which a network-arrival-time-based heuristic algorithm (NAT-HA) and a maximum-estimated-delay-based heuristic algorithm (MED-HA). The RL-based scheduler, first-in-first-out (FIFO), round-robin (RR), NAT-HA, and MED-HA were compared through large-scale simulations on four network topologies. The simulation results in fixed-packet generation scenarios showed that our proposal, the RL-based scheduler, achieved the minimization of maximum E2E latency in all the topologies. In other scenarios with random flow generation, the RL-based scheduler and MED-HA showed the lowest maximum E2E latency for all topologies. Depending on the topology, the maximum E2E latency of NAT-HA was equal to or larger than that of the RL-based scheduler. In terms of fairness, the RL-based scheduler showed a higher level of fairness than that of FIFO and RR. NAT-HA had similar or lower fairness than the RL-based scheduler depending on the topology, and MED-HA had the same level of fairness as the RL-based scheduler.

**Keywords:** min-max criterion; reinforcement learning; double-deep Q-learning; prioritized experience replay; end-to-end latency; fairness

## 1. Introduction

Fairness is an important ingredient in the consideration of many network problems, such as packet scheduling, congestion control, and bandwidth allocation. In terms of quality of service (QoS), fairness has been considered along with QoS in order to satisfy QoS for as many flows as possible with limited resources. Since it is highly correlated with QoS in these circumstances, the maximization of fairness has been extensively studied [1,2].

One approach recently proposed to enhance fairness employed the concept of metadata that describes the information from other data. It is used for scheduling and routing by adding metadata to packets, particularly in segment routing (SR) [3], deterministic networks (DetNet) [4], and so on. SR-based load balancers achieved better fairness while showing lower mean response time compared to random load balancers [5]. In Joung's study [6], a jitter upper bound guarantee technique using timestamp metadata for DetNet service was proposed, thereby increasing the quality of experience (QoE) fairness. It used real-time transport protocol (RTP) packets to use timestamp fields to guarantee jitter upper bound

through timestamps and buffers. In the DetNet data plane [7], it is specified that metadata such as timestamps can be exchanged between applications, so it is possible to guarantee the upper limit of the jitter in the DetNet environment even if RTP packets are not used. Some studies have used the remaining hop counts as metadata to improve fairness and performance. Wang's study [8] conducted routing and packet scheduling through weights calculated as remaining hop counts in a Network-on-Chip (NoC) environment. In Wang's study [8], packets were divided into packets with small remaining hop counts (PSR) and packets with large remaining hop counts (PLR). Accordingly, preferentially transferring packets with small remaining hop counts (PPSR) and preferentially transferring packets with large remaining hop counts (PPLR) were proposed as routing and packet scheduling methods. PPSR is good in terms of average end-to-end (E2E) latency to transmit PSR first, but fairness is low. Conversely, PPLR has good fairness but low performance. Additionally, an adaptive remaining hop count (ARHC) flow control method for switching between PPSR and PPLR was proposed. As a result of the simulation, ARHC showed better performance than PPLR, and fairness better than PPSR and round-robin, and better performance when the network is over-saturated. Another example of using metadata is programmable packet scheduling. In the case of programmable packet scheduling, it is sometimes assumed that the packet has metadata [9–12]. The packet's metadata can be in the payload or in the header. When adding metadata to the packet header, even if no fields are added to the header, metadata can be added while maintaining the existing protocol header structure by using information that can be inferred or unused fields [13,14].

Recently, studies have been proposed using reinforcement learning (RL) for such problems. Existing optimization or control algorithms in the current network environment, where a large volume and variety of traffic is serviced, have a disadvantage in that they are difficult to implement and apply in practice due to their high computational complexity. On the other hand, reinforcement learning can be applied to various environments because it can learn and infer the network environment or parameters by itself using observed raw data. Furthermore, since it is easy to incorporate the behavior of the agent and resource observation into a reinforcement learning problem in a network environment, many studies have attempted to solve network problems using reinforcement learning. Chen's study [15] used the actor-critic RL to optimize scheduling and video QoE in a wireless environment, and showed better performance in fairness than traditional algorithms and other learning algorithms. López-Sánchez's study [16] showed that the proposed deep RL-based scheduling is superior to other schedulers in terms of fairness and average latency. Although not in a network environment, Jiang's study [17] showed excellent fairness and efficiency in various problems through hierarchical RL and decentralized RL in a multi-agent environment. In addition to these, many studies are underway that apply RL and deep learning in a network environment [18,19]. Studies on packet scheduling with RL in the field of networks have also been proposed. In Kim's study [20], Q-learning was applied to design a scheduler that is robust to changes in data characteristics and satisfies the data latency requirements. Guo's study [21] used the deep Q-network for resource allocation for packet scheduling and satisfied the quality of service (QoS) of more packets than round-robin and priority-based schedulers.

Among studies that increase fairness, some studies increase latency fairness [22–24]. These studies are intended to solve the problem of low deadline achievement rate due to physical distance by considering the number of remaining hops of the packet. For example, the hop base multi-queue (HBMQ) proposed in [22] is a scheduler that puts packets into different queues for each number of remaining hops and provides a round-robin service. As a result, the service of packets passing over a long distance is guaranteed and the deadline achievement rate is increased.

However, the objective of past studies is mainly focused on the increase in latency fairness incidentally to equalize the deadline achievement rate, rather than on the increase in the fairness of E2E latency itself. Even if the deadline is met, users may experience different qualities of service between flows. In the current network, it is common to experience

service loss as the distance between the user and the server increases. However, in terms of QoE, this cannot be considered fair. Therefore, latency fairness must be considered even after the flow meets the deadline. In services such as metaverse, which provide constant latency regardless of the user's physical location, the fairness of latency itself must be considered beyond the level where latency fairness satisfies the deadline. To provide maximally fair latency to all users, latency fairness should be increased based on users experiencing maximum latency. Therefore, we propose a scheduler that increases latency fairness by minimizing the maximum E2E latency.

In this paper, we use an RL approach to minimize the maximum E2E latency for fairness in terms of latency experienced by packets from different paths. We note that the flow path is fixed, unlike the setting in which flow routing is taken into account described in Wang's study [8]. In addition, packet scheduling is performed by considering not only the remaining hop count but also the experienced delay up to the present. In Wang's study [8], the packet scheduler is a probability-based method calculated with weights by changing the weight calculation method, but the proposed method uses an RL approach, so a more flexible response is plausible. We proposed a method for estimating the E2E latency of a packet at a node by using two metadata: the number of remaining hops and the generation time of the packet. This allows the node to calculate the estimated delay of the packet. It was demonstrated that this estimated delay can be used as the state and reward in the RL-based scheduler to increase fairness and reduce the maximum E2E latency. By implementing the RL-based scheduler not only in a single-agent environment but also in a multi-agent environment with a node as an agent, it was shown that fairness can be increased even in a multi-hop environment using the RL-based scheduler. When the packet generation scenario is fixed, the RL-based scheduler always achieves the minimization of the maximum E2E latency. In addition, it was shown through simulations that the RL-based scheduler achieved the minimization of maximum E2E latency in various topologies. It showed that the maximum E2E latency is lower than that of first-in-first-out (FIFO) and round-robin (RR) schedulers even in an environment where random flows are generated. In addition, we proposed two heuristic algorithms for comparison with the RL-based scheduler using the metadata described above. Unlike scheduling algorithms that rely on the number of hops, all of these algorithms avoid starvation and do not heavily rely on probabilities for service decisions, resulting in superior performance. In simulations, the proposed heuristic algorithms outperformed FIFO or RR, with one of them achieving similar performance to the RL-based scheduler.

The structure of this paper is as follows. After the introduction in Section 1, we present the material and method in Section 2. We first review the RL algorithms used in this work. Afterwards, we describe the experimental settings, including an RL model for the simulations and the network topologies designed for the simulation environment. Additionally, we look at the heuristic algorithms proposed for comparison of the results. In Section 3, we show the results of the RL-based scheduler in the network topologies as a simulation environment we designed. We conclude the paper in Section 4 with a discussion and contribution drawn from the results.

## 2. Materials and Methods

### 2.1. Reinforcement Learning Algorithms

Reinforcement learning provides a computational framework for learning and decision-making from past experiences. The primary aim of RL is to find a policy to achieve a goal which maximizes a cumulative reward. It is usually accomplished by a cycle of 'perceive-reason-act' in the environment. Given the current state of the environment, an agent takes an action against the environment. This brings about a new state and returns a reward in due course. As the environment changes, the agent perceives the new states and repeats this process until it finds the optimal policy to achieve a goal. In this process, it requires a variety of experiences composed of state–action pairs through "*exploration*". Depending on the outcomes, the agent takes an action yielding the highest reward based on the agent's

experiences, called "*exploitation*". In doing so, the agent builds state value, or state–action value, as a means to find an optimal policy in the environment. Q-learning is a representative off-policy RL algorithm for agents to learn the policy. Given the state-action values, the agent selects the action that has the highest Q-value [25]. Q-value can be obtained through Equation (1).

$$Q(s_t,\ a_t) = (1-\alpha)Q(s_t,a_t) + \alpha\left(r_t + \gamma\max_a Q(s_{t+1},a)\right).\tag{1}$$

All mathematical symbols in Equation (1) are shown in Table 1.

**Table 1.** Symbols used in formulas.

| Symbol | Meaning |
| --- | --- |
| Q | Q-table or deep learning network |
| S, s | State |
| A, a | Action |
| R, r | Reward |
| T | Time |
| $\alpha$ | Learning rate |
| $\gamma$ | Discount factor |
| $\theta$ | Deep learning network's weights |
| Y | Target Q-value |

The Q-value represents the maximum cumulative reward achievable from a given state and action pair. The Q-value is updated to reflect the reward obtained immediately, weighted by the learning rate and the maximum cumulative reward that can be obtained in the future. The discount factor determines the proportion of future rewards to be considered, compared to the reward obtained at the present time. It is an exclusive value between 0 and 1, such that the greater the reward obtained in the distant future, the less it is reflected in the current Q-value. The policy determines the action that achieves the maximum Q-value, as given by Equation (1). Q-learning can learn the optimal policy in a finite Markov decision process. Because the tabular version of Q-learning in the environment lacks the ability to deal with large state- and action-spaces, deep Q-networks (DQN) was proposed which combines Q-learning and deep-learning [26]. As deep learning showed a remarkable capacity to generalize the value over a certain part of a state(-action) space, it enables the scaling down of the problem to a manageable size. However, DQN is likely to overestimate the problem in which the Q-value of a specific action continues to increase. This overestimating problem stems from the bias caused by approximating the state–action value using the max operator. Additionally, due to the max operator, selecting and evaluating an action using the same value is prone to overestimation. To prevent this, a double-deep Q-network (DDQN) that combines double Q-learning and DQN was introduced [27]. DDQN uses two separated Q-networks: one for the action selection and the other for the evaluation [28]. It is expressed as Equation (2).

$$Y_t = R_{t+1} + \gamma Q\left(S_{t+1}, \arg\max_a Q(S_{t+1},a;\theta_t), \theta_t^-\right).\tag{2}$$

All mathematical symbols in Equation (2) are also shown in Table 1.

Equation (2) shows that there are two Q-networks, $\theta$ and $\theta^-$. DDQN obtains the Q-value using Equation (1) with $Y_t$ calculated using Equation (2), and updates the theta parameter of the Q-network that determines the action. Although $\theta$ changes during learning, the $\theta^-$ parameter of the Q-network used to calculate the future Q-value remains constant, allowing the Q-value to change for each episode by adjusting only the currently obtained reward. Consequently, the expected cumulative future reward changes minimally, making it less prone to falling into local optimization. To use the Q-network learned so far for future Q-value calculations, $\theta^-$ is periodically updated by copying the current

$\theta$ parameter. The policy determines the action to be taken by obtaining the maximum Q-value using the Q-network with $\theta^-$. DDQN showed more stable training than DQN by effectively reducing the overestimate problem, and therefore exhibited better performance than DQN [27].

Deep RL, an integration of RL algorithm and deep learning, requires a buffer to hold experiences. In the present learning method, experiences were randomly chosen from the buffer, and trained the deep RL afterwards. Prioritized experience replay (PER) is an experience replay buffer designed to learn by the choice of novel experiences rather than randomly chosen ones from the buffer [29]. PER obtains the TD-error for each sample and sets the priority for each sample. Equations (3) and (4) are used for calculating TD-error and priority in the DDQN model, respectively:

$$\text{TD error} = |Y_t - Q(S_t, A_t, \theta)|, \tag{3}$$

$$\text{priority} = (TD\ error + \epsilon)^{\beta} \tag{4}$$

where $\beta$ is a variable between 0–1 that determines the level at which priority is determined by *TD error*, and $\epsilon$ is a very small constant added so that priority does not become 0. Deep RLs using PER showed better performance than cases where PER was not used because it selects high-priority samples to proceed with learning [29].

Practical domains that can be encountered in real environments, such as autonomous driving, robots, and games, are environments where multiple agents influence each other. RL in which multiple agents compete or cooperate to solve problems in such an environment is called multi-agent RL (MARL). Conversely, RL with one agent is called single-agent RL (SARL). The simplest MARL is independent Q-learning (IQL): a set of multiple Q-learning agents in the environment where each agent acts on its own policy independently based on its observed states [30]. IQL does not perform well when there are many agents or when there are various types of agents, but it works well in a simple environment.

### 2.2. Problem Statement

We consider an output buffered switch in Figure 1, where each output port module has a buffer. Table 2 explains the symbols used in Figure 1 and the simulation environment. The buffer of the output port module contains the output queues based on input ports. Each queue is paired with an input port to store packets coming from the corresponding input port. To implement such a node, the packet must contain input port information. The total number of queues in a node is M × N. The scheduler of the output port module uses an RL-based scheduler. The scheduler of nodes not using the RL model operates as a FIFO.

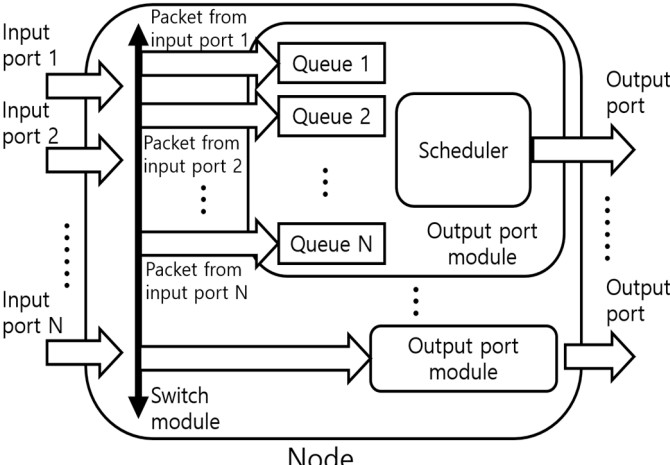

**Figure 1.** Node architecture used in the simulation.

**Table 2.** Symbols used in topology description.

| Symbol | Meaning |
|---|---|
| M | Number of an output port |
| N | Number of an input port |
| H | Number of nodes |
| C | Link capacity |
| T | Unit time in simulation |

Let us consider a network that consists of three nodes with the topology shown in Figure 2. The sources in the network generate packets according to the packet generation scenarios in Table 3. T refers to the unit time of the simulation. We assume that the transmission delay occurring on the link in the simulation always takes T for every packet. This is to make it easier to observe the results by making all transmissions and receptions occurring in the simulation performed every T. To fix the transmission delay to T, link capacity is fixed to C, and the length of all packets is fixed to C × T. It is assumed that all traffic originating from the source is of the same type, of the same nature, and of the same priority, since competing packets must have the same characteristics for fair competition. It is also assumed that the propagation delays of all links have a negligibly small value. Therefore, the propagation delay does not affect the E2E latency, so the E2E latency is obtained in units of T. As the simulation operates in units of T, the packet generation time points are also determined in units of T in the packet generation scenario. Since the scheduler is only meaningful when packets in the node need to be sent to the same port simultaneously, the packet creation scenario was designed with this consideration in mind. Table 3 presents a scenario in which packets contend once at node 0 and once at node 1. Although there may be no contention at node 1, the scheduler at node 0 selects a packet that increases the maximum end-to-end delay time, distinguishing it from the optimal case. Therefore, additional packets are not generated for contention at node 1.

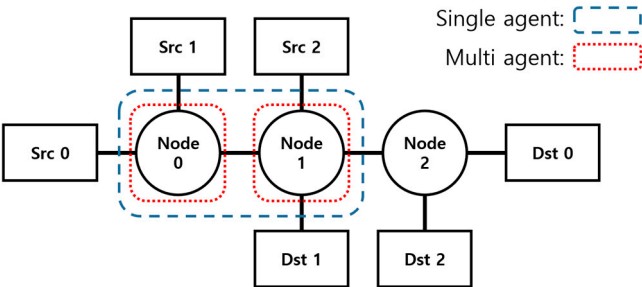

**Figure 2.** Example topology.

**Table 3.** Packet generation scenario in the example topology.

| Time | Source 0 | Source 1 | Source 2 |
|---|---|---|---|
| 0 | 1 | 1 | |
| T | | | 1 |

Since such a network environment requires precise time synchronization, it is well suited to a time-sensitive network, especially a LAN environment [31]. In this environment, the lowest possible maximum E2E latency in the example topology is 3T. To minimize the maximum latency, the following service scenario must be executed: At time 0 in node 0, when two packets from the two sources want to go out on the same link, the packet generated at source 0 has to travel a larger number of hops and must be sent out first. At time T, when two packets in node 1 from the two input ports want to go out on the same link, the two packets have the same number of remaining hops, but the packet generated

from source 0 has already suffered a delay time T, so it must be sent out first. When the scheduler operates in this way, the E2E latencies of packets generated from sources 0, 1, and 2 are 3T, 2T, and 2T, respectively, and the maximum E2E latency is minimized. The maximum E2E latency that occurs when the packet generated from source 0 is not transmitted first is 4T. In this case, the average E2E latency may be smaller than the average E2E latency when the maximum E2E latency is 3T. However, since we are interested only in the minimization of the maximum E2E latency, the average E2E latency will not be used as a performance indicator.

As in the example topology, in the fixed packet generation scenario, since the maximum E2E latency for each scheduler is within a fixed range, performance can be compared only with the maximum E2E latency. However, in random packet generation scenarios, different performance evaluation indicators are needed because the maximum observable E2E latency is different for each scenario. For quantitative performance evaluation, latency fairness and maximum latency fairness are evaluated using Jain's fairness index [32]. The fairness index is obtained using (5).

$$\text{Fairness index} = \frac{\left(\sum_{i=1}^{n} x_i\right)^2}{n \sum_{i=1}^{n} x_i^2},\tag{5}$$

In (5), $x_i$ is the E2E latency of packet *i* and n is the number of packets. For maximum latency fairness, $x_i$ is the maximum E2E latency of packets within flow *i* and n is the number of flows. The value of the fairness index is between 0 and 1. If the maximum E2E latency over all the packets is minimized, then the E2E latencies of other packets inevitably increase, so latency fairness increases. Therefore, the quantitative evaluation of performance is possible through the latency fairness index.

When using SARL in the example topology, the states that the agent can observe are the states of node 0 and node 1. In SARL, there is a virtual controller connected to these nodes, receiving state information from the nodes, and based on this, the controller determines the operation of the node's scheduler. The controller notifies each node of the decided action, and the node operates the scheduler accordingly. In the simulation, the delay due to the exchange of information between the node and the controller is not taken into account. When using MARL in the example topology, the agent implements an agent on node 0 and node 1. The MARL method uses the IQL. IQL generally has poor performance, but in the assumed network topology, all agents are identical and operate sequentially, so IQL can show sufficient performance because there is little influence on each other. Each agent can only observe the state of its node. In MARL, the node and the agent are the same, so the scheduler can be operated through the RL model without exchanging information between the agent and the node. In MARL, agents cannot directly observe E2E latency. Therefore, the agent minimizes the maximum value of the estimated delay that can be calculated by the node without minimizing the maximum E2E latency. Let the *i*th packet in the queue be called $p_i$, $i \geq 1$. If the number of remaining hops of $p_i$ is $p_i^h$, the generated time of $p_i$ is $p_i^g$, and the length of $p_i$ is $p_i^l$, the estimated delay of $p_i$ can be obtained using (6).

$$p_i's \text{ estimated delay} = \left(p_i^h - 1\right) \times \left(\frac{p_i^l}{C}\right) + \text{current time} - p_i^g + \sum_{k=1}^{i} p_k^l / C,\tag{6}$$

In order not to calculate the transmission delay redundantly, the remaining number of hops is used by subtracting 1. The estimated delay is the expected minimum E2E latency of $p_i$, where there is only one queue. To calculate (6) in the node, the packet must contain information about the generated time and the number of remaining hops as metadata. Both metadata have been used in previous studies [8,16,22], but either one of them was used or it was calculated and used at the node without being recorded in the packet. As shown in Equation (6), the estimated delay can be reflected by matching the units of the two metadata,

so it is not governed by specific metadata. Since the estimated delay is consequently the same as the E2E latency, if the maximum value of the estimated delay is minimized, the maximum E2E latency is minimized. SARL can achieve E2E latency by adding information exchange between the controller and the destination node so that the agent can obtain the state information of the destination node. However, since the burden on the controller is increased and the E2E latency is only obtained when the packet reaches its destination, SARL also uses estimated delay. Estimated delay is calculated at every step at every node. The estimated delay reflects the queuing delay, so it either maintains or increases as the step increases. Since the estimated delay information changes steadily at every step, the RL agent uses the estimated delay information together with the queue information.

*2.3. SARL and MARL*

RL models perceive the state information from the environment and select the action based on that. Through actions, agents obtain rewards and change states. By repeating this, the RL model learned an optimal policy, a series of actions that maximizes the expected amount of reward. To solve the problem of minimizing the maximum E2E latency of the network using RL, the state, action, and reward are defined as follows.

- State

The state uses information that can be obtained from the queue for each output port. For each queue, the remaining hop count of the first packet, the delay to the present of the first packet, the queue length, and the maximum estimated delay of the packets in the queue are obtained. The state is information from which the maximum E2E latency can be inferred. The number of remaining hops and the delay to the presence of the first packet are information for calculating the estimated delay of the first packet. The first packet is important because the estimated delay and the actual E2E latency are almost similar. The queue length can infer the degree to which it affects other queues. The maximum estimated delay of packets in the queue is the closest value to the target maximum E2E latency.

In MARL, the state space size is $4 \times M \times N$. In SARL, the state space size is $4 \times M \times N \times H$. In the example topology, node 1 has 2 input ports and 2 output ports, so the state space size in MARL is 16. However, if only one of the output ports wants to send packets out at the same time, the other output ports may use FIFO. Therefore, the state space size can be reduced to 8 by using only the information of the queue of one output port. SARL can likewise reduce the state space size from 32 to 16. In a fixed topology and scenario, the state space size can be reduced in this way to reduce the computational cost of the RL model, but in a non-stationary environment, all information must be used, so the state space size cannot be reduced.

- Action

The action is to select a queue to service per output port. The action is performed when both transmission and reception are completed. Since the number of queues per output port is equal to the number of input ports, the action space size in MARL is $N \times M$. In SARL, the action space size is $N \times M \times H$. The action space size can be reduced just like the state space size. In the example topology, with MARL, the action space size of node 1 is 4, but the valid number of the output port is 1, so the action space size can be reduced to 2. Similarly, in SARL, the action space size can be reduced from 8 to 4.

- Reward

The reward is computed using Algorithm 1. The agent receives the reward when all transmissions and receptions of the action are completed once the action is performed for each output port. When a packet is transmitted, it is compensated using the estimated delay of the transmitted packet and the maximum estimated delay of the packet in the queue after all transmission and reception are completed. The key to Algorithm 1 is that the larger the estimated delay of the transmitted packet is and the smaller the maximum estimated delay of the remaining packets is in the queue, the greater the compensation

is. If a packet is not sent even though there are packets in the queue to send, a penalty is given with the amount of the square of the maximum expected latency of the packets in the queue.

In SARL, all rewards of nodes are summed up and used as the actual reward, so the penalty obtained by not sending from one node should be able to affect the entire network. Assuming that the estimated delay of one packet is n, the maximum reward that can be given from this packet in SARL is $n^2$. For this reason, it is beneficial to operate by work conserving in all nodes by receiving a penalty of $n^2$ when the transmission is not performed, so the penalty is calculated as a square. The rewards obtained for each output port in MARL are added up and used as the actual reward. In SARL, all the rewards obtained for each output port for each node are added up and used as the actual reward. In MARL, the reward of a single node is directly used, so the penalty can be set smaller.

---

**Algorithm 1.** Rewards in a single node.

---

M = number of the output port
N = number of the input portOutputPort
Module[1..M]
OutputPortModule.Queue[1..N]
reward = 0

for i in 1..M:
    if all OutputPortModule[i].Queue is not empty
        if packet transmission is complete:
            if all queues are empty:
                reward += estimated delay of the transmitted packet
            else:
                reward += estimated delay of the transmitted packet −
                    max(estimated delay of packets in all queues)
        else:
            reward += −max(estimated delay of packets in all queues)$^2$
return reward

---

In SARL, the environment has a structure shown in Figure 3. The controller is physically connected to the nodes in the network, receives status information, and sends control messages to each node. In Figure 3, the action list means a control message. Since the queue to be serviced in the actual node must be determined for each output port, it is expressed as an action list. In this process, a transmission delay occurs. However, considering this transmission delay, the time unit of the event in the simulation cannot be fixed to T. To set the simulation time unit to T, the transmission delay of the control message can also be fixed to T, or it can be implemented so that the control message contains not only the action to be performed now but also the action to be performed at a more distant future time. Although this transmission delay is not negligible, it is a delay caused by a structural problem, not a delay caused by the performance of the scheduler. In the actual simulation, we wanted to see the maximum E2E latency difference due to the scheduler, so the transmission delay between the controller and the node was ignored. Additionally, the controller combines the state of each node and uses it as a single state. In the example topology, it is assumed that all events occur in units of T, so all nodes send state information to the controller at the same time. It is assumed that the processing delay occurring in the controller is small enough to be negligible.

MARL uses the IQL method, so the node information is all of the states that the agent can observe, as shown in Figure 4. If one node is drawn including the RL model, it is shown in Figure 5. In Figure 5, the dotted line indicates logical division. In other words, there is no physical distinction between the RL model and the scheduler, and it is implemented in all nodes, so MARL does not need to consider transmission delay, unlike SARL. State information obtained from each output port is combined and used as the actual state,

which is the same for SARL. In Figure 5, the state information of all output port modules is received, but as described above, the size of state and action can be reduced by not receiving the state information of some output port modules. As with SARL, it is assumed that the processing delay occurring in the RL model is small enough to be negligible.

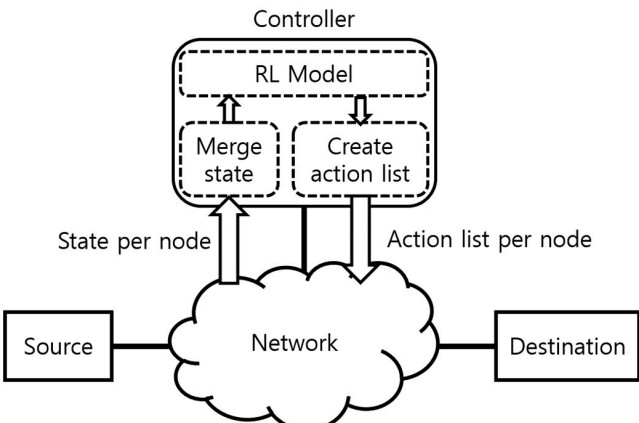

**Figure 3.** Full environment structure in SARL.

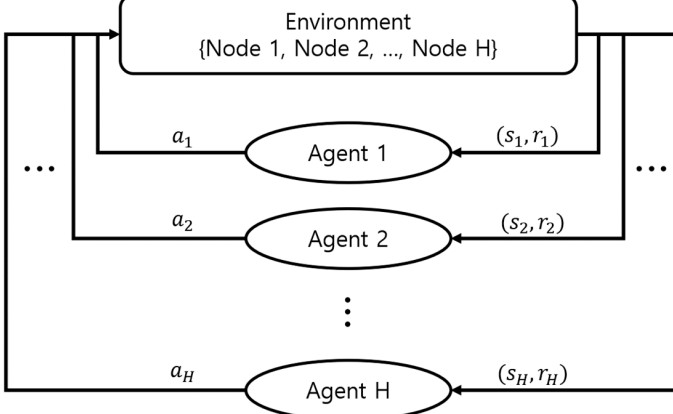

**Figure 4.** The learning process of MARL in the system.

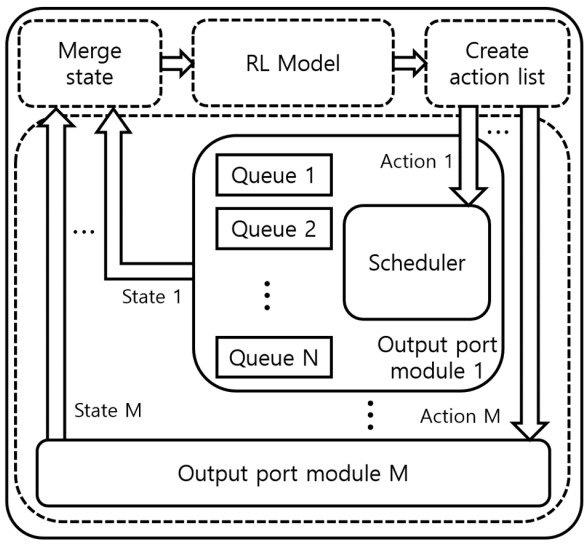

**Figure 5.** A single node structure in MARL.

### 2.4. Scheduler Using Metadata

It is difficult to compare the performance of existing schedulers that increase latency fairness because they have different problems to solve. For example, in the case of PPLR [8], the maximum E2E latency cannot be fixed because it is probability-based transmission, and in HBMQ [22], it is difficult to see a difference in performance from RR because nodes in the assumed environment consist of queues for each input port. Largest delay first (LDF) [16] determines the service order by finding the sum of the queuing delays of all packets in the queue at the node without using the number of hops information. Unlike PPLR [8], less probability is applied and the method is robust against starvation because the queue to be served is determined using latency information. However, since the number of hops information is not used, packets passing through different routes cannot be serviced. In addition, since the node calculates and uses the queuing delay, it does not have information on the delay time that the packet experienced before, and thus its performance deteriorates in a multi-hop environment. For comparison with the RL-based scheduler, we propose two heuristic algorithms that can achieve the minimization of maximum E2E latency.

The first heuristic algorithm is a network-arrival-time-based heuristic algorithm (NAT-HA). Algorithm 2 explains the operation of NAT-HA. NAT in Algorithm 2 stands for network arrival time. By recording the network arrival time in the packet as metadata, the queue with the smallest network arrival time of the head of the queue (HoQ) is serviced. If the network arrival time of the HoQ is the same, service is provided randomly among these queues. The network arrival time indicates the time spent in the network. Since the E2E latency increases as much as the time spent in the network, the packet that has stayed in the network for a long time, that is, the packet that entered the network first, is served first to reduce the maximum E2E latency in a single network. NAT-HA has the same performance as FIFO in a single-node network. Even in a network with multiple nodes, the operation of the ingress node is the same as that of FIFO. Therefore, the worst-case performance of NAT-HA is the same as the worst-case performance of FIFO. In terms of average performance, NAT-HA shows better performance than FIFO because it can provide services in the direction of reducing E2E latency in the core node. NAT-HA is similar to LDF in that it records the queuing delay in packets as metadata. The difference lies in the method used to estimate the delay time. NAT-HA estimates the delay time up to the present using the packet creation time, whereas LDF continuously adds queuing delay at the node. The delay time that the packet has experienced so far calculated by the node differs depending on the path of the packet, allowing for the reflection of the difference based on the number of hops. However, not all nodes can know this information, so a heuristic algorithm that uses the number of hops along with the delay time is also required.

---

**Algorithm 2.** NAT-HA in the output port module.

---

```
M = number of the input port
Queue[1..M]
min_NAT = Inf
idx = 0

while true
    for i in 1..M:
        if Queue[i] is not empty
            if min_NAT > Queue[i].head.NAT
                idx = i
                min_NAT = Queue[i].head.NAT
    send(Queue[idx].head)
    Queue[idx].dequeue
```

---

The second heuristic algorithm is a scheduling algorithm that uses the estimated delay in the same way as the RL-based scheduler. The metadata also uses the remaining hop count

and packet-generated time in the same way. This algorithm calculates the estimated delay of all packets in the queue and serves the queue containing the packet with the maximum estimated delay. This is called the maximum-estimated-delay-based heuristic algorithm (MED-HA). The method for obtaining estimated delay and maximum estimated delay in MED-HA is described in Algorithm 3. If there are several packets with the maximum estimated delay, the queue is serviced randomly among the queues with them. As mentioned earlier, the estimated delay is the minimum E2E latency that the packet can achieve. The actual E2E latency varies depending on the situation of other queues or nodes that will pass in the future, but in a situation where a packet that determines the maximum E2E latency is observed, the estimated delay is equal to the maximum E2E latency. Additionally, the estimated delay can better estimate the E2E latency at the node just before the destination. Therefore, MED-HA can perform better than FIFO or RR in terms of maximum E2E latency. Additionally, unlike [8], MED-HA does not use only the remaining hop count but also uses the packet-generated time, so it is effective to reduce the maximum E2E latency because packets have different weights even in the same remaining hop count situation. In addition, [8] used probability-based scheduling to prevent starvation, but MED-HA prevented starvation by increasing the estimated delay when packets were not sent. By the same logic, unlike NAT-HA, MED-HA considers the number of remaining hops, so that when the network arrival times of packets in a single network are the same, the MED-HA can service the packet more efficiently. MED-HA has better performance than NAT-HA, but it uses a larger amount of metadata and requires a process of calculating the estimated delay of all packets.

---

**Algorithm 3.** MED-HA in the output port module.

---

C = link capacity
M = number of the input port
Queue[1..M]
max_estimated_delay = 0
idx = 0
bit_length = 0

while true
    for i in 1..M:
        if Queue[i] is not empty
          for j in 1..Queue[i].length
            estimated_delay = Queue[i][j].remaining_hop_count
                $\times$ (Queue[i][j].length/C)
                + time.now $-$ Queue[i][j].generated_time
                + (bit_length/C)
          bit_length += Queue[i][j].length
          if max_estimated_delay < estimated_delay
            idx = i
            max_estimated_delay = estimated_delay
    send(Queue[idx].head)
    Queue[idx].dequeue

---

We demonstrate how the two heuristic algorithms work in the example topology. Figure 6 represents the output port module of node 0 at time 0. Since source 0 and source 1 send packets to node 0 at the same time, the network arrival time and generated time of these packets are the same. NAT-HA provides a uniformly random service among these queues because the network arrival time of each HoQ is the same. MED-HA first calculates the estimated delay of all packets. Since the current time is 0, both packets have experienced a delay of 0, and since both packets are HoQ, their order in the queue is 0. After calculating the estimated delay, the estimated delay of the packet in queue 0 is 3T, and the estimated delay of the packet in queue 1 is 2T. Since packets with the maximum estimated delay are in queue 0, MED-HA serves queue 0.

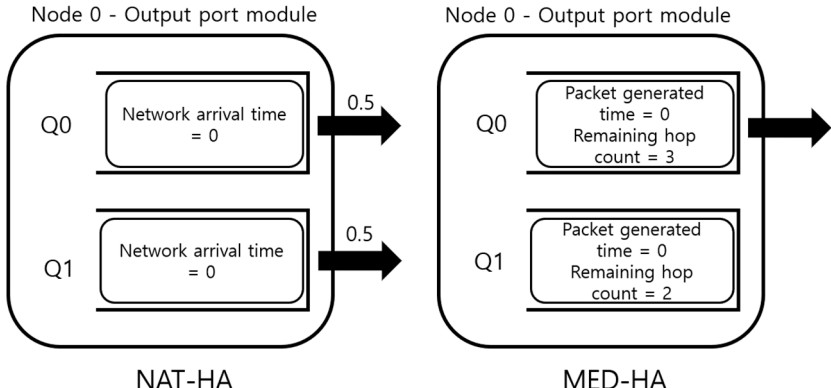

**Figure 6.** In the example topology, when packets want to go out simultaneously from node 0 at time 0, the service of each heuristic algorithm is displayed.

Figure 7 expresses the output port module when the packet of source 0 and the packet of source 2 want to go out from node 1 at the time T. Since the network arrival time of HoQ of queue 0 is smaller than the network arrival time of HoQ of queue 1, NAT-HA serves queue 0. MED-HA calculates the estimated delay of the packets. Although the remaining hop counts of the two packets are the same, the experienced delay up to the present is different because the packet generation time is different. As a result of calculating the estimated delay, the estimated delay of the packet in queue 0 is 3T, and the estimated delay of the packet in queue 1 is 2T. Since packets with maximum estimated delay are in queue 0, MED-HA serves queue 0. In the example topology, NAT-HA can minimize the maximum E2E latency with a probability of 50% depending on which packet goes first at time 0. In contrast, MED-HA always minimizes the maximum E2E latency.

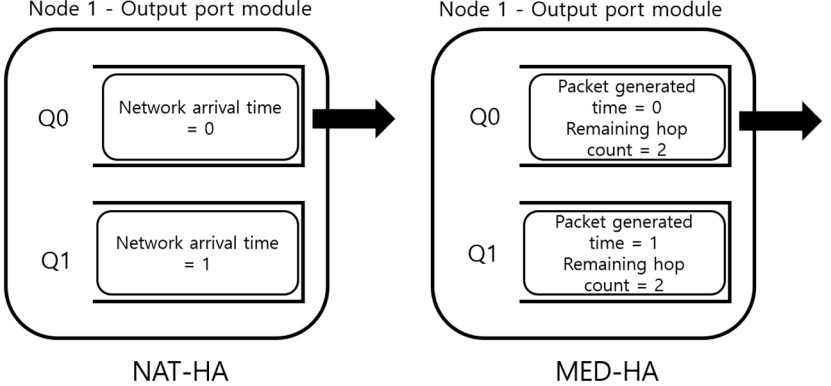

**Figure 7.** In the example topology, when packets want to go out simultaneously from node 1 at time T, the service of each heuristic algorithm is displayed.

## 3. Results

### 3.1. Simulation Setup

The simulation environment has the same characteristics as the environment described in the example topology and operates in units of T. The simulation is designed to create a scenario where packets want to go out from the node to the same port at the same time, using 2 × 2 nodes as in the example topology. In the 2 × 2 node, there is one queue for each input port at the output port. Schedulers other than FIFO determine the queue to receive service from among the two queues for each T. To exclude the possibility of packet loss in the simulation, the capacity of the queue is assumed to be infinite. Since the maximum delay time increases only when packets undergo maximum queuing delay in the queue, packet loss is not considered in the simulation.

The RL model was implemented using TensorFlow. TensorFlow is an E2E open source platform for building machine learning models [33]. The structure of the deep neural

network used in the simulation is shown in Table 4. The Dense layer is a layer that connects all inputs and outputs. Rectified linear unit (ReLu) is a type of activation function and is a nonlinear function, as shown in (7).

$$f(x) = \max(0, x). \tag{7}$$

**Table 4.** Deep learning network architecture.

| Type | Size |
|---|---|
| Input | The state size |
| Dense | 64 |
| ReLu | 64 |
| Dense | 64 |
| ReLu | 64 |
| Dense(Output) | 1 |

The size of the Q-network was set to the smallest size that was found to be optimal through simulations. Although there would be no problem in learning even with larger hidden layers, the size of the Q-network had to be considered within the limited resources of the nodes in the MARL environment, as it must be implemented on them. The size of the Q-network can vary depending on the environment, and for the random packet generation scenario, the size was increased to $256 \times 256$ to accommodate the increased complexity of the environment. The RL buffer uses PER and has a size of 50,000. Table 5 shows the hyperparameters of the RL model. The learning rate of the RL model decreases according to the cosine decay method, and the cosine decay follows (8). The d and $\alpha_0$ are hyperparameters that determine the period during which the learning rate decays.

$$\text{Learning rate} = \alpha_0 \times 0.5 \times \left( 1 + \cos\left( \pi \times \frac{\min(\text{number of episodes}, \quad d)}{d} \right) \right), \tag{8}$$

**Table 5.** Hyperparameters.

| Hyperparameter | Value |
|---|---|
| Number of episodes | 3000 |
| Initial learning rate ($\alpha_0$) | 0.002 |
| Discount factor | 0.99 |
| Initial epsilon | 1 |
| Epsilon decay | 0.997 |
| Minimum epsilon | 0.001 |
| Target model update cycle | 500 episodes |
| Batch size | 256 |
| Learning rate decay period (d) | 2500 episodes |

The reason for reducing the learning rate is to keep the global optimum from deviating due to the high learning rate.

The episode starts at time 0 and ends when all packets have reached their destination or time 100T. The reason for ending an episode at a specific point in time is that the proposed reinforcement learning scheduler may select an empty queue, preventing packets from being sent out. The end time of 100T was chosen, taking into consideration the topology, number of generated packets, and their timing. A total of 3000 episodes were repeated per simulation. During the first 500 episodes of the simulation, the RL model does not train and stores samples in a buffer. After that, the RL model is trained for the remaining 2500 episodes. Among the hyperparameters, epsilon decay and learning rate decay period were determined considering the number of learning episodes. Other hyperparameters were determined empirically while conducting simulations. The simulation was repeated 15 times for each topology by dividing it into SARL and MARL, and the results were obtained.

When the simulation begins, packets are generated from the source and transmitted to the node based on the packet generation scenario. At each time step, the scheduler uses the observed state to decide which queue to serve. Information about the state, action, next state, and reward is obtained and stored as a sample in the buffer at each step. During the first 500 episodes, no training is performed, and the epsilon value is set to 1, so the queue to be served is chosen randomly. After 500 episodes, exploration and exploitation are performed using epsilon to determine the queue to serve and store the sample. Learning occurs at the end of each episode after 500 episodes. The simulation ends after 2500 episodes, and the Q-network parameters are saved. The test results are obtained by averaging the results of one or several episodes of models that have completed their training.

### 3.2. Simulation

The first simulation was conducted in topology 1, as shown in Figure 8. Topology 1 generates packets according to Table 6. Topology 1 is an extension of the example topology, and the packet generated from source 0 must be transmitted first to minimize the maximum E2E latency. The minimum value of the maximum E2E latency of topology 1 obtained in the same way as in the example topology is 5T. To compare the performance of the RL-based scheduler, we scheduled using FIFO, RR, NAT-HA, and MED-HA. In addition to topology 1, topology 2, topology 3, and topology 4, which will be described later, we compared the performance of these four schedulers. The maximum E2E latency and latency fairness index were used as performance indicators.

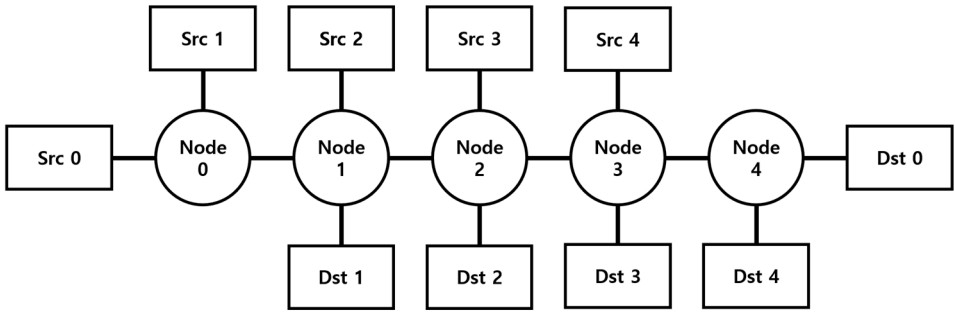

**Figure 8.** Topology 1.

**Table 6.** Packet generation scenario in topology 1.

| Time | Source 0 | Source 1 | Source 2 | Source 3 | Source 4 |
|------|----------|----------|----------|----------|----------|
| 0 | 1 | 1 | | | |
| T | | | 1 | | |
| 2T | | | | 1 | |
| 3T | | | | | 1 |

Figure 9 is a graph of the learning results of the RL-based scheduler in topology 1. It can be seen that both SARL and MARL achieve minimization of the maximum E2E latency. During the first 500 episodes, the RL model does not train, so the sum of rewards does not increase or the maximum E2E latency decreases. After 500 episodes, as the RL model starts learning, the sum of rewards increases and reaches the maximum value, and the maximum E2E latency is minimized. Looking at the MARL graph, it can be seen that even if the moving average graph of the sum of rewards changes, the moving average graph of the maximum E2E latency does not change. This means that the maximum E2E latency can be minimized even if the RL model does not reach the global optimum. However, when the RL model reaches the global optimum, the maximum E2E latency is always minimized.

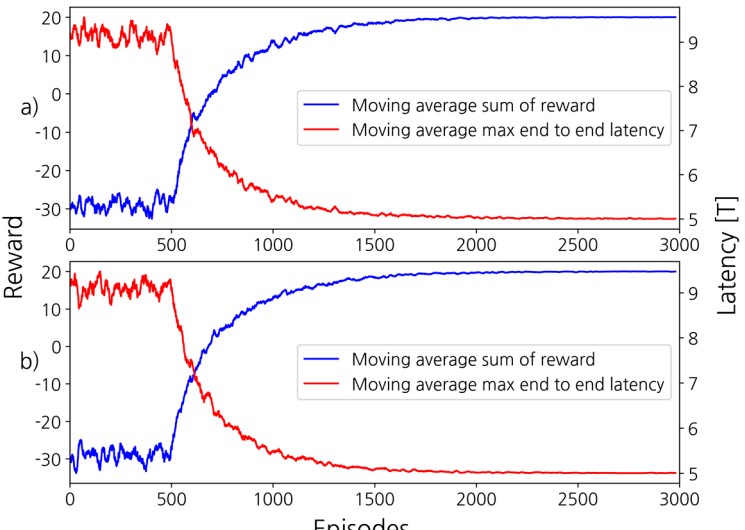

**Figure 9.** Training result of RL model in topology 1. (**a**): SARL, (**b**): MARL.

The second simulation was performed in topology 2 as shown in Figure 10. Topology 2 generates packets according to Table 7. Topology 2 is a tree-shaped topology and an environment where E2E latency can diverge due to increased queuing delay caused by bottleneck links. In this environment, the RL model can learn from more diverse queue situations than topology 1 due to the presence of bottleneck links.

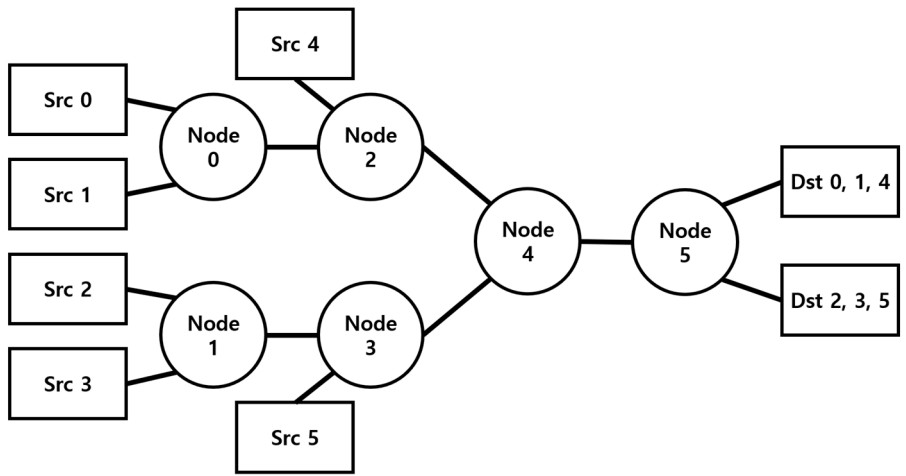

**Figure 10.** Topology 2.

**Table 7.** Packet generation scenario in topology 2.

| Time | Source 0 | Source 1 | Source 2 | Source 3 | Source 4 | Source 5 |
|------|----------|----------|----------|----------|----------|----------|
| 0    | 1        | 1        | 1        | 1        |          |          |
| T    |          |          |          |          | 1        | 1        |

Figure 11 is a graph of the training results of the RL-based scheduler in topology 2. Both SARL and MARL achieved minimization of the maximum E2E latency. In the case of SARL, it can be confirmed that the local optimum was reached and maintained during the first 1000 episodes of training. This phenomenon is caused by the stability of RL algorithms and deep learning networks. This is because the state space size and the action space size in the SARL of topology 2 are large compared to the deep learning network structure.

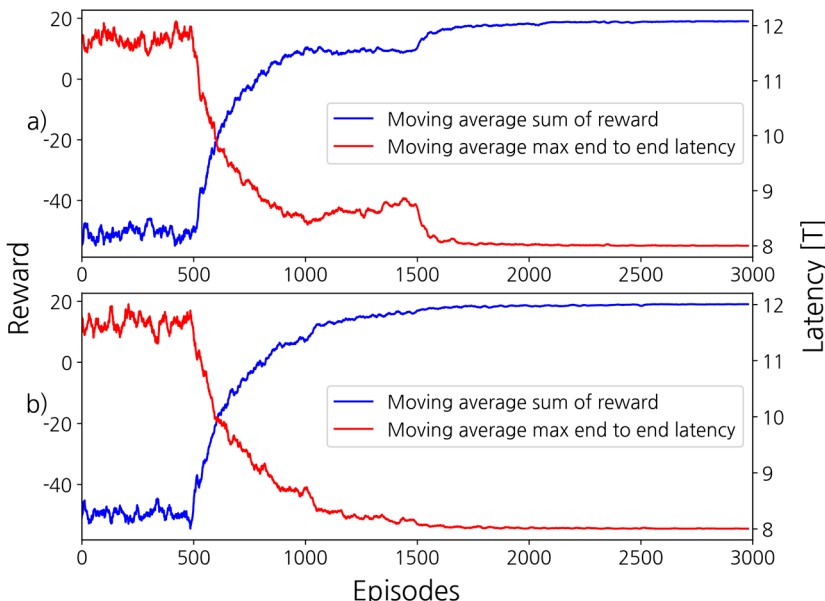

**Figure 11.** Training result of RL model in topology 2. (**a**): SARL, (**b**): MARL.

To see this clearly, a simulation of sending more packets in topology 2 was performed. In the packet generation scenario, 10 packets were randomly generated, but the maximum length of the scenario was 20T.

Figure 12 shows the performance comparison between the DQN algorithm and the DDQN algorithm in this environment and the performance difference according to the deep learning network structure. Unlike previous simulations, more packets are randomly generated, resulting in more varied states. In such an environment, the DQN algorithm converges to a lower optimum than DDQN regardless of the deep learning network structure. This is because the stability of DQN is lower than that of DDQN due to the overestimation problem, which is a limitation of DQN. Additionally, the performance of each algorithm depends on the deep learning network structure. This is because the state space size and the action space size in the SARL of topology 2 are large compared to the deep learning network structure, and the states that occur due to the random flow are diverse. In SARL, the state space size and action space size are determined by the number of input ports, the number of output ports, and the number of nodes. In particular, the action space size increases in proportion to the square of the number of output ports and the number of nodes. Therefore, in a topology with a large number of nodes, it is necessary to increase the size of the deep learning network structure to learn more stably. If the state is not varied, it falls into the local optimum for a short time as shown in Figure 11. However, when the state is diversified, the stability of the deep learning network structure is directly related to the performance. In Figure 12, when the size of the deep learning network structure increases from 64 × 64 to 256 × 256, both DQN and DDQN algorithms find a better optimum.

The third simulation was performed in topology 3 as shown in Figure 13. Topology 3 generates packets according to Table 8. Unlike the topologies discussed above, topology 3 assumes that a delay has already been experienced when generating a packet. It is assumed that packets generated by sources 0, 1, and 2 have the experienced delay of T, 7T, and 3T, respectively. Topology 3 can achieve the minimization of the maximum E2E latency by emptying the queue to immediately send out the packet, which determines the maximum E2E latency. Since topology 3 is a topology consisting of one node, the implementation of SARL and MARL is not different.

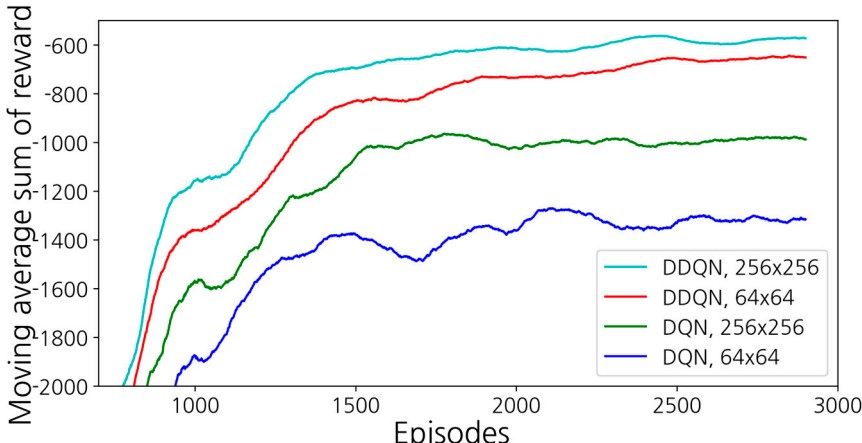

**Figure 12.** Training results according to SARL's RL algorithm and deep learning network structure when random flows are generated in topology 2.

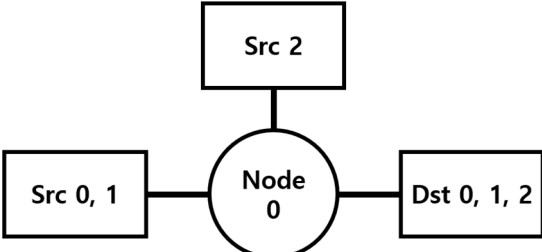

**Figure 13.** Topology 3.

**Table 8.** Packet generation scenario in topology 3.

| Time | Source 0 | Source 1 | Source 2 |
|------|----------|----------|----------|
| 0 | 1 | | 1 |
| T | | 1 | |

Figure 14 is a graph of the learning results of the RL-based scheduler in topology 3. RL achieves the minimization of maximum E2E latency. In Figure 14, it can be seen that the sum of rewards and the maximum E2E latency change rapidly around episode 1000. This is because topology 3 is a small topology consisting of one node, and the time slot required for simulation is also small, so the state is not varied. In such a topology, the cumulative reward and maximum E2E latency can vary significantly with one action, as shown in Figure 14.

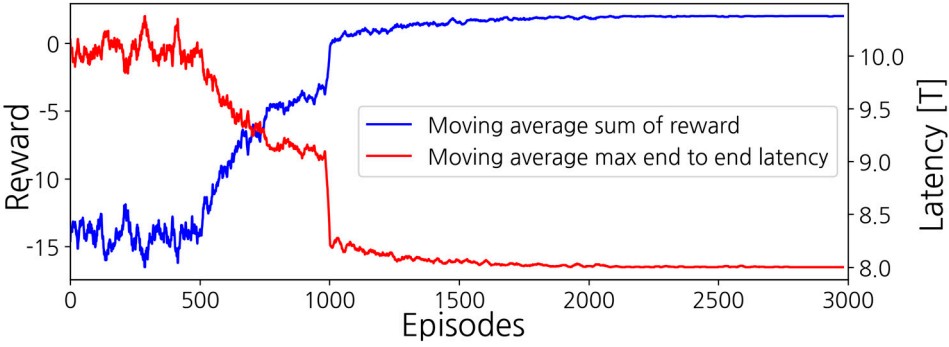

**Figure 14.** Training result of RL model in topology 3.

According to the previous simulation, it was confirmed that the proposed RL method works well in the fixed scenario. Unlike previous simulations, this simulation examines the performance of the RL scheduler in an environment that generates random flows. A random flow generates 10 packets in a packet generation scenario of up to 20T in length. In an environment that generates a random flow, the number of packets passing through the network increases and the simulation takes longer, so the training episodes were increased from 3000 to 10,000. The hyperparameters were also modified accordingly. Unlike the existing topologies, the topology examines the performance of the RL scheduler in a topology where a symmetric cycle exists. The topology used for the simulation is shown in Figure 15. Table 9 shows the flow path along Topology 4. Topology 4 is a topology with cycles, where the scheduler's choice can affect itself in the future. In this topology, we wanted to determine if scheduling is possible with the RL-based scheduler considering the possible future impact. Topology 4 generates random flows because it is not possible to construct a scenario such that a particular flow overlaps all other flows. This simulation implements only MARL among RL-based schedulers. Topology 4 has 9 nodes, 2 input ports, and 2 output ports, respectively, and the action space size in SARL is $2^{(2 \times 18)}$, which is too large to be implemented. On the other hand, MARL can be implemented because the action space size is 4 and the state space size is 16.

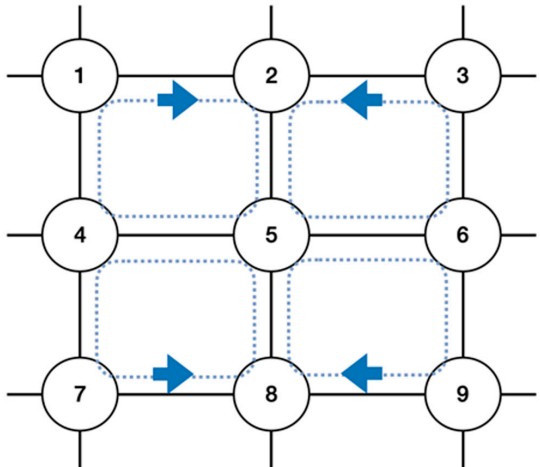

**Figure 15.** Topology 4 [34].

**Table 9.** The routes of the flow in topology 4.

| Flow Number | Route |
| --- | --- |
| 0 | 1-2-5-6-9 |
| 1 | 4-1-2 |
| 2 | 4-7-8 |
| 3 | 7-8-5-6-3 |
| 4 | 3-2-5-4-7 |
| 5 | 6-3-2 |
| 6 | 6-9-8 |
| 7 | 9-8-5-4-1 |

Figure 16 shows the moving average of the maximum E2E latency of each scheduler in a network environment where random flows are generated in topology 4. Both MED-HA and MARL showed the best performance, followed by NAT-HA, RR, and FIFO. In topology 4, RR performs better than FIFO because the number of remaining hops for packets that want to go out at the same time may be different. In a fixed scenario, there is no performance difference between RR and FIFO because these cases cannot be continuously experienced. However, in an environment that creates a random flow, it is better to guarantee the service of the queue in this case in terms of maximum E2E latency.

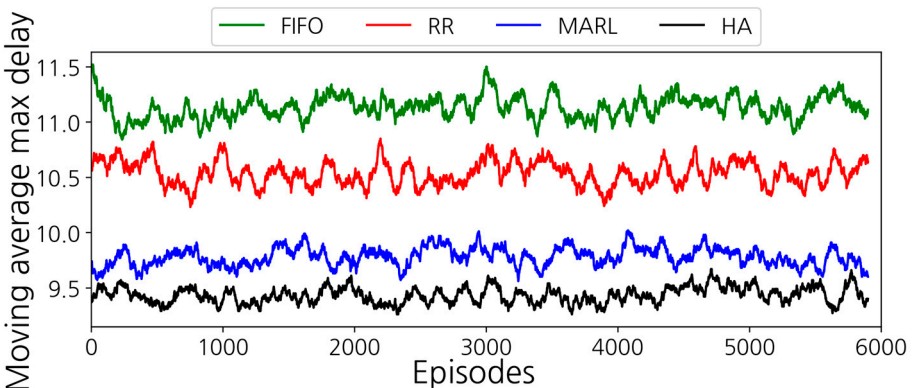

**Figure 16.** Moving average of maximum end-to-end latency per scheduler when random flows are generated in topology 4.

In topology 4, Figure 17 and Table 10 show the latency fairness index of packets per scheduler. The fairness indexes obtained (5) were represented with %. The latency fairness index was calculated for each episode, and the episode repeated a total of 10,000 times. The average latency fairness performance of each scheduler was, in order, MED-HA, MARL, NAT-HA, FIFO, and RR. The RL-based scheduler and MED-HA had about 4% higher latency fairness than other schedulers, and were good in terms of upper and lower bounds. In the case of NAT-HA, maximum E2E latency decreased, but the latency fairness index did not increase significantly compared to FIFO and RR. NAT-HA shows an average latency fairness index similar to that of FIFO and RR. NAT-HA behaves similarly to RR in topology 4. Since only two flows compete with each other for each output port in each node, if the service is based on the arrival time, there is a high probability that the service will be provided like RR. However, unlike RR, one queue can be continuously serviced, and the ingress node of a flow like node 1 often prefers to service a specific flow, so the average latency fairness index is higher than that of RR. For the same reason, FIFO also behaves like RR, and the average latency fairness index of RR and NAT-HA is similar, but FIFO obtains the advantage that NAT-HA obtains from an ingress node probabilistically, so the average latency fairness index is lower than that of NAT-HA.

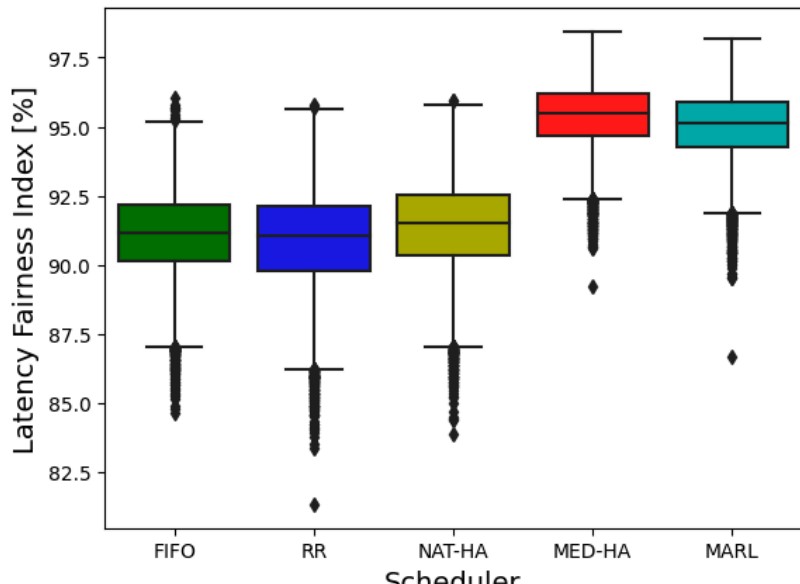

**Figure 17.** Box plot of latency fairness index per scheduler from 10,000 episodes in topology 4 with the random flow.

**Table 10.** Latency fairness in topology 4 with random flow.

|  | Latency Fairness Index [%] | | | | |
|---|---|---|---|---|---|
|  | **FIFO** | **RR** | **NAT-HA** | **MED-HA** | **MARL** |
| Mean | 91.1 | 90.89 | 91.38 | 95.39 | 95.01 |
| Min | 84.62 | 81.33 | 84 | 89.2 | 96.65 |
| Max | 96.07 | 95.78 | 96.01 | 98.44 | 98.18 |

Figure 18 and Table 11 show the results of calculating the maximum latency fairness index for each episode for 10,000 episodes. As shown in Figure 18, maximum latency fairness results show good performance in the same order as the latency fairness results. MED-HA and RL-based schedulers had 2–3% higher average maximum latency fairness than other schedulers and have a higher lower bound. Through this, it was shown that the proposed RL-based scheduler and MED-HA reduced the maximum E2E latency while increasing the fairness of all latencies and the fairness of the maximum latency between flows. RR had the worst performance in terms of latency fairness index and maximum fairness index. In terms of latency fairness, the latency fairness index increases when the difference between latencies is small. However, since RR is transmitted as specified regardless of the order in which it entered the node or network, the deviation between latencies inevitably increases. In terms of maximum latency fairness, RR can have a large deviation in the maximum E2E latency for each flow, so the maximum latency fairness index is also low. However, depending on the topology, RR can be better in terms of latency fairness and maximum latency fairness. For example, when burst traffic enters an ingress node sequentially, the fairness index of RR will be higher than that of FIFO.

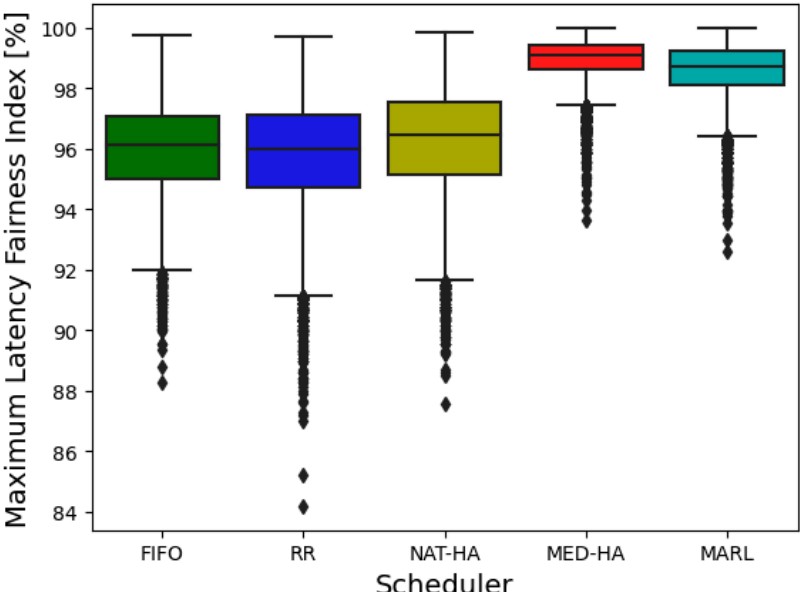

**Figure 18.** Box plot of maximum latency fairness index per scheduler from 10,000 episodes in topology 4 with the random flow.

**Table 11.** Maximum latency fairness in topology 4 with random flow.

|  | Maximum Latency Fairness Index [%] | | | | |
|---|---|---|---|---|---|
|  | **FIFO** | **RR** | **NAT-HA** | **MED-HA** | **MARL** |
| Mean | 95.96 | 95.79 | 96.22 | 98.91 | 98.57 |
| Min | 88.28 | 84.18 | 87.55 | 93.64 | 92.6 |
| Max | 99.78 | 99.73 | 99.82 | 100 | 100 |

In order to examine the performance of SARL in an environment where random flows are generated, additional simulations were performed in topology 2. Since this simulation is also an environment that generates a random flow, the number of training episodes was increased to 10,000, and the hyperparameters were modified accordingly. Because the state is diversified due to the random flow, we changed some of the deep learning network structure and parameters so that the RL-based scheduler could learn well. SARL changed the deep learning network structure to (state space size × 256 × 256 × 256 × action space size) while maintaining the layer type. The MARL environment maintains the existing deep learning network structure because the state does not vary much due to random flow.

Figure 19 shows the moving average of the maximum E2E latency of each scheduler in a network environment where random flows are generated in Topology 2. The proposed RL-based scheduler outperformed FIFO and RR and showed similar performance to MED-HA and NAT-HA. In topology 2, even in the fixed scenario, the RL-based scheduler showed the same performance as MED-HA and NAT-HA, so it can be said that SARL and MARL reached the global optimum. In topology 2, unlike topology 4, RR performed worse than FIFO. The reason is that, contrary to topology 4, the remaining hop count of packets is always the same when they want to go out at the same time. In such an environment, it is not good to guarantee the service of queues like RR, as the latency experienced by the network determines the maximum E2E latency.

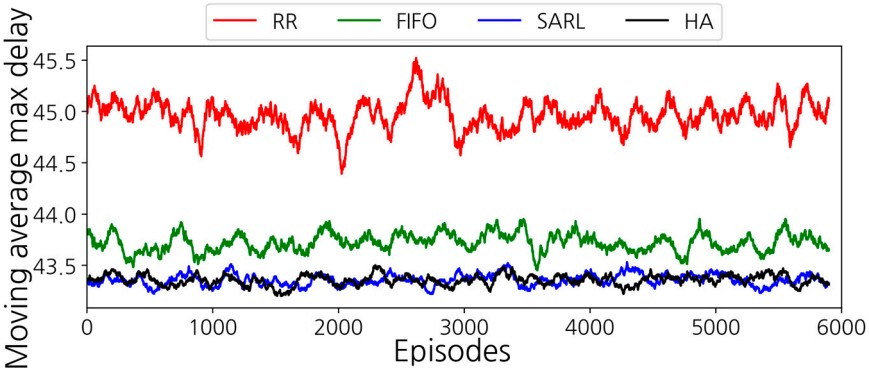

**Figure 19.** Moving average of maximum E2E latency per scheduler when random flows are generated in topology 2.

Figure 20 and Table 12 show the latency fairness index of each scheduler in % units. The latency fairness index was obtained for 10,000 episodes for each episode. In terms of average latency fairness, NAT-HA was the best, followed by SARL, MED-HA, MARL, FIFO, and RR in order of performance. As with the maximum E2E latency results, MED-HA and RL-based schedulers had similar latency fairness indices. However, NAT-HA produced better results than MED-HA and the RL-based scheduler because topology 2 is an environment in which E2E latency diverges and packets accumulate in queues. In topology 2, only packets with the same number of remaining hops content, so there is no difference in the maximum E2E latency of NAT-HA or other proposed schedulers. If each queue has a packet with the maximum estimated delay, MED-HA or an RL-based scheduler serves random queues. However, at this time, the estimated delay of HoQ may be different. NAT-HA is advantageous in terms of latency fairness because it determines the order of packet service considering only HoQ. Compared to the proposed schedulers, FIFO and RR have an average latency fairness index of 0.5% and 2% smaller, respectively. The reason why the difference with FIFO is small is that there is no large performance difference from the scheduler proposed by the node connected to the bottleneck link for the same reason as NAT-HA. However, the average latency fairness index is smaller than that of the proposed scheduler due to performance differences in other nodes.

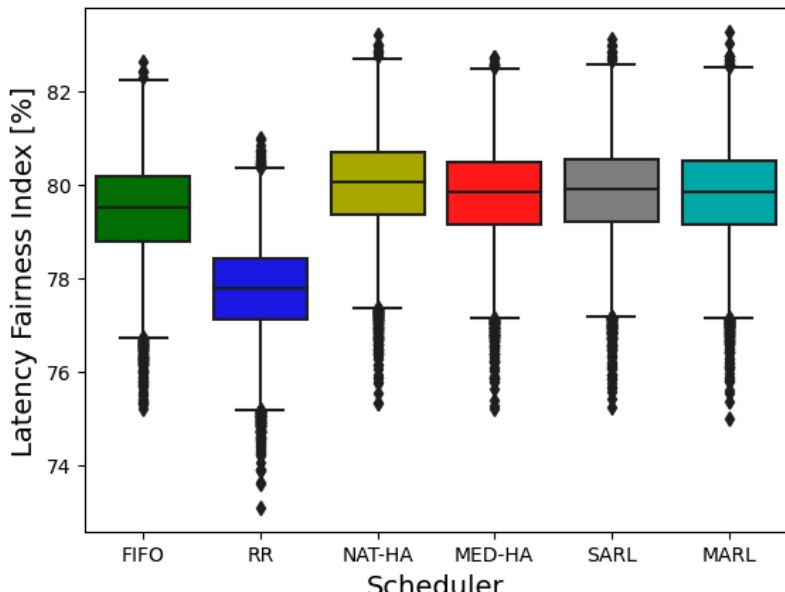

**Figure 20.** Box plot of latency fairness index per scheduler from 10,000 episodes in topology 2 with the random flow.

**Table 12.** Latency fairness in topology 2 with random flow.

| | Latency Fairness Index [%] | | | | | |
|------|-------|-------|--------|--------|-------|-------|
| | **FIFO** | **RR** | **NAT-HA** | **MED-HA** | **SARL** | **MARL** |
| Mean | 79.46 | 77.75 | 80 | 79.8 | 79.85 | 79.81 |
| Min | 75.21 | 73.1 | 75.33 | 75.22 | 75.24 | 75 |
| Max | 82.62 | 81.01 | 83.21 | 82.73 | 83.13 | 83.28 |

Figure 21 and Table 13 are the results of calculating the maximum latency fairness index for each episode for 10,000 episodes. Looking at the maximum latency fairness index in Figure 21, unlike the latency fairness index, there is no difference in average maximum latency fairness between the proposed algorithms. This is the same reason why there is no difference in the maximum E2E latency described above. FIFO, like latency fairness, has an average maximum latency fairness index that is about 0.5% lower. RR is about 4.5% lower than the average maximum latency fairness index. This is because the bottleneck link in topology 2 exists right before the destination node, so RR increases the maximum E2E latency. Through the simulation in topology 2 with the random flow, it was confirmed that the proposed algorithms reduce the maximum E2E latency and increase the latency fairness and maximum latency fairness event in an environment where divergence occurs in a bottleneck link.

Table 14 shows the average maximum E2E latency for each simulation of each scheduler. The RL-based scheduler minimizes the maximum E2E latency in all simulations. NAT-HA achieves maximum E2E latency minimization in topology 2 regardless of how packets are generated. MED-HA achieved the minimization of the maximum E2E latency in all other simulations except topology 3. Additionally, looking at Table 14, it can be seen that the proposed heuristic algorithms perform better than FIFO and RR even in fixed scenarios. This is because the metadata used by the heuristic algorithm is information that can better estimate the E2E latency, so there is a high probability of transmitting a packet with a large E2E latency. The RL-based scheduler shows the same performance as these heuristic algorithms, but shows better performance in one topology. This is because the RL-based scheduler can perform scheduling by considering the future state in a fixed scenario. There is no maximum E2E latency difference between SARL and MARL in all topologies. MARL has the advantage that it can be applied in topology on a larger scale

than SARL, but it is expected that the performance will decrease due to the limitations of the IQL method as the scale increases. Compared to SARL, MARL can be applied on a larger scale in topology and has the advantage of being able to perform better with a smaller deep learning network. However, as the scale increases, performance is expected to decrease due to the limitations of the IQL method.

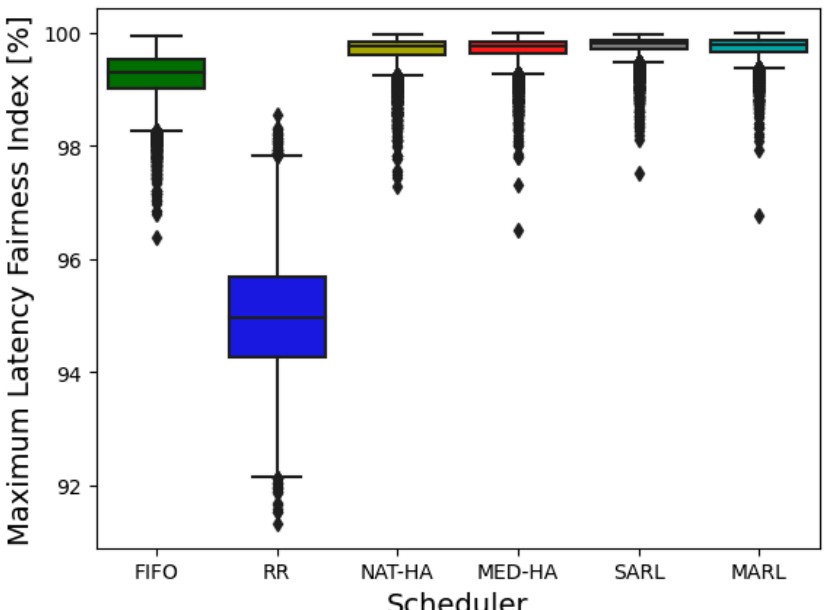

**Figure 21.** Box plot of maximum latency fairness index per scheduler from 10,000 episodes in topology 2 with the random flow.

**Table 13.** Maximum latency fairness in topology 2 with the random flow.

| | Maximum Latency Fairness Index [%] | | | | | |
| | **FIFO** | **RR** | **NAT-HA** | **MED-HA** | **SARL** | **MARL** |
|---|---|---|---|---|---|---|
| Mean | 99.22 | 94.99 | 99.68 | 99.7 | 99.77 | 99.74 |
| Min | 96.38 | 91.33 | 96.49 | 96.5 | 97.51 | 96.76 |
| Max | 99.93 | 98.54 | 99.98 | 99.99 | 99.97 | 99.99 |

**Table 14.** Comparison of average maximum E2E latency by the scheduler.

| Topology | Average Maximum E2E Latency | | | | | |
| | **FIFO** | **RR** | **NAT-HA** | **MED-HA** | **SARL** | **MARL** |
|---|---|---|---|---|---|---|
| 1 | 5.9375T | 5.9375T | 5.5T | 5T | 5T | 5T |
| 2 | 9T | 9T | 8T | 8T | 8T | 8T |
| 3 | 9T | 9T | 9T | 9T | 8T | 8T |
| 2 with random flow | 43.711T | 44.13T | 43.281T | 43.27T | 43.279T | 43.274T |
| 4 with random flow | 11.134T | 10.481T | 10.173T | 9.433T | | 9.423T |

## 4. Conclusions

In this paper, we proposed an RL-based scheduler that minimizes the maximum E2E latency and evaluated its performance with other schedulers through simulations. The RL model was implemented based on DDQN with PER structure using TensorFlow. In addition, for performance comparison, two heuristic algorithms that can minimize E2E latency, NAT-HA and MED-HA, were proposed. In the network topology, the node assumes that the output port module has as many queues as the number of input ports, and that the link always takes the transmission delay of T for every packet.

The RL agent is implemented in two ways: SARL and MARL. In SARL, the controller receives information and issues commands through message exchange with nodes as agents. The simulation did not take the delay into account due to these message exchanges. In MARL, the agent for each node makes an independent decision based on the information in the node, so there is no delay due to message exchange. In an RL environment, the states use information from the queues. MARL uses the queue information of all the output ports of a single node, whereas SARL uses the queue information of all the output ports of all the nodes as the state. In the RL environment, the agent selects a queue to serve each output port as an action. SARL selects a queue to be serviced by the output port for each node observed by the controller, which is the agent. In the RL environment, the reward was obtained through the estimated delay of transmitted and remaining packets. Estimated latency refers to the minimum E2E latency that a packet can currently achieve.

Simulations were performed in four topologies, and in topologies 2 and 4, random flows were generated and simulations were performed. FIFO, RR, NAT-HA, and MED-HA simulations were additionally performed for performance comparison with the RL-based scheduler. Simulation results showed that FIFO and RR did not achieve maximum E2E latency minimization in all topologies. NAT-HA achieved the minimization of the maximum E2E latency only in topology 2, and MED-HA achieved the minimization of the maximum E2E latency in topologies 1 and 3. The RL-based scheduler achieved the minimization of maximum E2E latency in both SARL and MARL in all topologies, which are fixed scenarios. In scenarios with random flow generation, the RL-based scheduler showed the lowest average maximum E2E latency for all topologies, the same for MED-HA. Depending on the topology, the average maximum E2E latency of NAT-HA was equal to or higher than that of the RL-based scheduler. In terms of fairness, the RL-based scheduler showed a 4% and 0.5% higher average latency fairness index than FIFO and 4% and 2% higher than RR, respectively, by topology. NAT-HA showed an average latency fairness index that was 4% lower than the RL-based scheduler in one topology and similar in the other topologies. MED-HA had the same average latency fairness index as the RL-based scheduler in all topologies. In addition, the RL-based scheduler and MED-HA showed a higher average maximum latency fairness index than other schedulers in all topologies.

The proposed RL-based scheduler is implemented using a value-based RL algorithm. However, since the value-based RL algorithm cannot be applied in an environment where the action space size is infinite, it is difficult to apply the proposed method as the topology grows. Therefore, we will propose an RL-based scheduler using policy-based RL, such as the proximal policy optimization algorithm, and investigate the performance of the RL scheduler in a large-scale network. Moreover, to account for a more realistic environment, we propose a RL model that enhances QoS fairness among high-priority traffic by either integrating it with an existing priority scheduling algorithm or extending the state to include more network parameters. In addition, we will propose an RL-based scheduler that can learn and operate in real time without operating in unit time by making various traffic characteristics such as packet length, burst size, and input rate, which can perform asynchronous learning. To this end, we propose an RL model that optimizes processing delay using computational complexity as an evaluation index, as well as a method for calculating estimated delay time considering processing delay.

**Author Contributions:** Conceptualization, J.K. and J.J.; methodology, J.K. and J.R.; software, J.K. and J.R.; validation, J.K., J.J. and J.H.L.; formal analysis, J.K.; investigation, J.K.; resources, J.K.; data curation, J.K.; writing—original draft preparation, J.K.; writing—review and editing, J.J. and J.H.L.; visualization, J.K.; supervision, J.J.; project administration, J.J. All authors have read and agreed to the published version of the manuscript.

**Funding:** This work was supported by the National Research Foundation of Korea (NRF) grant funded by the Korea government (MSIT, Ministry of Science and Information and Communication Technology) (No. 2020R1F1A1058591).

**Institutional Review Board Statement:** Not applicable.

**Informed Consent Statement:** Not applicable.

**Data Availability Statement:** Not applicable.

**Conflicts of Interest:** The authors declare no conflict of interest.

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
