# Peer review of "Improving End-To-End Latency Fairness Using a Reinforcement-Learning-Based Network Scheduler"

_applsci, doi:10.3390/app13063397_

Round 1

Reviewer 1 Report

Authors propose in this paper the reduction the end-to-end (E2E) latency using Reinforcement Learning  based scheduler. The paper describes a solution to a significant problem of end-to-end (E2E) latency on quality of service (QoS) regardless of the user’s physical location.  This is an interesting paper on a topic of much interest currently. However, there are a few aspects of the paper that I think the authors could have presented better.

1. The introduction provides a short discussion of the contribution with the existing literature, however, please provide a description of the contribution or clearly list the milestones that make the contribution and compare with the existing literature.

2. Perhaps in the introduction what is proposed above will be done and in a new section a discussion will be made comparing this paper and its contributions with the existing literature. That is, not only related to the use of RL, but also to other methods that have contributed to the reduction of latency in E2E.

3. The simulations make a comparison with other techniques or algorithms, however I think they should describe it as I suggest in 2, which really allows a discussion of the differences between them.

4. It is also possible to include other metrics for discussion, for example implementation, computational effort, etc. Perhaps not make those comparisons that could be very long, but at least have a discussion that allows to see the novelty and above all the interest of the paper proposal.

Reviewer 2 Report

The authors have written an article on end-to-end latency fairness using a reinforcement learning-based network schedule.

·        The article does not explain the simulation setup and assumptions. Include a subsection in the results section and provide a detailed explanation of the simulation environment.

·        In simulations, we get good results. But how does reinforcement learning perform in real-time network traffic?

·        Network topology, traffic patterns, and resource availability are crucial while designing an accurate model. Please elaborate in your article on how did you consider these parameters or any other aspects were considered? I did not find it in the submitted article.

·        Reinforcement learning algorithms typically require large amounts of training data to learn effective policies. In network scheduling, this can be difficult to obtain, as the performance of the scheduler may depend on factors that are difficult to measure or predict. Explain the process and approach of data collection and model training.

The article is well written, but simulation results cannot be completely agreed upon. The authors revise the manuscript and elaborate on the application of reinforcement learning in real time scenarios as well. Please revise the article according to the above point.

Reviewer 3 Report

The purpose of this research was to highlight one method for improving end-to-end latency fairness by employing a network scheduler that was based on reinforcement learning. Even though the work does not go too far from previously conducted research, I was still able to discern some valuable insights that were supplied by it due to the ambitious nature of the project. Nevertheless, there are still a few questions that need to be answered, and they are as follows:

1.       What is the difference between an “increase in latency fairness incidentally to equalize the deadline achievement rate, with an increase in the fairness of E2E latency”? This has not been greatly discussed and there is no foundational understanding provided by you, despite grounding your work as the “research gap” that you are willing to fill in. In addition, considering that other research investigations have used RL, why are you employing it as well?

2.       There is a need to provide bullet points on the contribution of the research in the second to last paragraph of section 1.

3.       Other minor issues in section 1 are as follows:

**** the maximization of fairness has been paid much attention **** rephrase this claims and describe further

***In Joung's work**** cite this at the instance

***Wang's study conducted **** cite this at the instance

***In Wang's study **** cite this at the instance

*** In Chen's study **** cite this at the instance

*** López-Sánchez's study**** cite this at the instance

** *Jiang's study**** cite this at the instance

*** In Kim's study**** cite this at the instance

***Guo's study **** cite this at the instance

** . In [8]**** cite the name associated with the number at the instance

***After the induction in Section I*** wrong spelling

4.       Section 2 is one of the good parts of the paper and is so detailed, but the authors do not highlight clearly the point/location where they have their scenario different from the other previous research studies.

5.       The double deep Q-network (DDQN) supposedly the main tool for this study was not explained in details, specifically equation 1 and 2 was not described by operations and are not in details

6.       The result of the work is quite interesting but not surprising, the network topology is description is ok, but the experimental scenario snapshot should be provided

Round 2

Reviewer 2 Report

the authors have addressed my comments.